# Contextual control of conditioned pain tolerance and endogenous analgesic systems

Sydney Trask[1], Jeffrey S Mogil[2], Fred J Helmstetter[3], Cheryl L Stucky[4], Katelyn E Sadler[4]*

[1]Department of Psychological Sciences, Purdue University, West Lafayette, United States; [2]Department of Psychology and Anesthesia, Alan Edwards Centre for Research on Pain, McGill University, Montreal, Canada; [3]Department of Psychology, University of Wisconsin-Milwaukee, Milwaukee, United States; [4]Department of Cell Biology, Neurobiology and Anatomy, Medical College of Wisconsin, Milwaukee, United States

*For correspondence: ksadler@mcw.edu

Competing interest: The authors declare that no competing interests exist.

**Abstract** The mechanisms underlying the transition from acute to chronic pain are unclear but may involve the persistence or strengthening of pain memories acquired in part through associative learning. Contextual cues, which comprise the environment in which events occur, were recently described as a critical regulator of pain memory; both male rodents and humans exhibit increased pain sensitivity in environments recently associated with a single painful experience. It is unknown, however, how repeated exposure to an acute painful unconditioned stimulus in a distinct context modifies pain sensitivity or the expectation of pain in that environment. To answer this question, we conditioned mice to associate distinct contexts with either repeated administration of a mild visceral pain stimulus (intraperitoneal injection of acetic acid) or vehicle injection over the course of 3 days. On the final day of experiments, animals received either an acid injection or vehicle injection prior to being placed into both contexts. In this way, contextual control of pain sensitivity and pain expectation could be tested respectively. When re-exposed to the noxious stimulus in a familiar environment, both male and female mice exhibited context-dependent conditioned analgesia, a phenomenon mediated by endogenous opioid signaling. However, when expecting the presentation of a painful stimulus in a given context, males exhibited conditioned hypersensitivity whereas females exhibited endogenous opioid-mediated conditioned analgesia. These results are evidence that pain perception and engagement of endogenous opioid systems can be modified through their psychological association with environmental cues. Successful determination of the brain circuits involved in this sexually dimorphic anticipatory response may allow for the manipulation of pain memories, which may contribute to the development of chronic pain states.

## Editor's evaluation

This study covers a series of experiments designed to characterize conditioned pain processing using a novel animal model in which mechanical nociception (von Frey test) and writhing are assessed following exposure to contextual cues that have been paired with visceral pain (intraperitoneal acetic acid injection). The results reveal that such cues exert complex, dose- and sex-dependent effects on pain processing. These experiments address an important topic from a translational perspective, because learning is an important but understudied contributor to the human pain experience and because there is evidence for sex differences in human pain expression. These findings will be of broad interest to researchers across fields of associative learning, neuroscience, and pain research.

## Introduction

Chronic pain development may involve the generalization and strengthening of acute pain memories (*Moseley and Vlaeyen, 2015*). Neuronal encoding of painful stimuli, known as nociception, is an evolutionarily conserved, unconditioned response (UCR) generated upon exposure to a tissue-damaging unconditioned stimulus (UCS). When pain persists, subjects are either repeatedly presented with or chronically experiencing this UCS, providing opportunity for various external cues or conditioned stimuli (CS) to become associated with the UCS. Pairings of the painful UCS with external cues can eventually lead to the development of a conditioned response (CR) elicited by the CS alone. In this way, nociceptive signaling and resulting pain behaviors may be modulated by previously biologically irrelevant stimuli.

*Pavlov, 2010*, was the first to describe the process through which a biologically irrelevant stimulus could come to exert control over physiological processes in a way that allows the organism to prepare for the upcoming disruption of homeostasis produced by the UCS. In addition to auditory stimuli like those used in Pavlov's experiments, the immediate environment or 'context' may control these preparatory responses as was demonstrated in an important experiment performed by *Siegel et al., 1982*. Over the course of 15 days, Siegel et al. treated rats with escalating doses of a mu-opioid receptor agonist (heroin) in one distinct physical environment. After training, animals received an ostensibly lethal dose of heroin (96.4% mortality rate in heroin-naïve rats) in the same environment where training occurred, or in a separate, novel environment. When administered in a novel context, this dose of heroin killed 64.3% of animals; the same amount of heroin administered in the familiar context decreased the death rate by ~50% such that only 32.4% of animals died. Similar to the CR first reported by *Pavlov, 2010*, this work suggested that the familiar environment served as a CS that predicted upcoming heroin administration, thereby initiating internal processes associated with drug tolerance. This phenomenon has since been termed a 'conditioned compensatory response' as exposure to the CS begins the process of compensating for the upcoming drug effect (see also *Siegel, 1977*; see *Siegel et al., 2000*, for a review). Here, we used a similar experimental paradigm to determine if repeated exposure to an acute painful stimulus in a specific context affects sensitivity to that stimulus. In other words, do animals exhibit conditioned compensatory responses to painful stimuli? And if they do, what is the physiological basis of that response?

It is also well documented that context can exert control over learned behaviors related to fear and anxiety (*Bouton and Bolles, 1979*) as well as over instrumental and choice behaviors (*Trask and Bouton, 2014*). Recently, context was found to influence pain behaviors in male mice and humans after a single conditioning trial (*Martin et al., 2019*). Both male mice and men exhibited stress-induced hyperalgesia if tested in an environment previously associated with a painful stimulus (*Martin et al., 2019*). Similarly, in a place avoidance task, male mice avoided the environment previously associated with this same noxious UCS (*Martin et al., 2019*). Neither female mice nor women exhibited conditioned hypersensitivity, conditioned place aversion, or context-dependent increases in circulating stress hormones in these experiments, suggesting that context did not impact nociceptive signaling in females as it did in males. However, an alternative interpretation of these data may be that contextual cues engaged compensatory endogenous analgesic systems in females, even in the absence of ongoing pain, such that their pain behaviors pre- and post-conditioning were indistinguishable. This hypothesis is appealing given the potential therapeutic implications that could follow if environment is capable of modulating endogenous analgesic mechanisms. To date, the few studies that have examined environmental influence on conditioned analgesia have primarily done so in fear conditioning paradigms. In general, those studies found that following fear conditioning in which a CS predicts an aversive shock outcome, animals exhibited increased pain tolerance (i.e., conditioned hypoalgesia) in the environment, or context, where fear conditioning occurred (*Fanselow and Helmstetter, 1988*; *Harris and Westbrook, 1994*; *Harris and Westbrook, 1995*; *Helmstetter and Fanselow, 1987*; *Watkins et al., 1993*). Although the conditioned fear responses exhibited following this associative learning have been extensively studied, the potential compensatory analgesic responses produced by context exposure have received little to no attention outside of these paradigms. We therefore designed a novel set of experiments to determine if context could be used as a CS to engage endogenous analgesic mechanisms and thus increase pain tolerance, in the absence of overt fear conditioning or ongoing pain.

# Results

## Female mice develop context-dependent analgesia after training with ascending doses of acetic acid

In chronic pain conditions and models, subjects are repeatedly or continuously presented with a noxious UCS. In these studies, we used associative learning paradigms that repeatedly coupled a painful UCS (i.e., acetic acid) with a unique environment to determine if pain memories affect pain sensitivity (UCR) and pain expectation (CR). To first determine if context can exert conditional control over pain sensitivity, we designed a within-subject, 3-day conditioning paradigm in which animals were trained to associate one unique context with escalating doses of intraperitoneal (IP) acetic acid and a second context with IP vehicle administration (*Figure 1A*). Prior to being placed into either chamber on the fourth day of the paradigm, animals were injected with a dose of acetic acid that was higher than the doses previously administered in the acid-trained context. Two pain behaviors were assessed in these experiments: (1) mechanical sensitivity of the hindpaw and (2) abdominal writhing. Although the hindpaw tissue is not directly injured by the UCS, the sensitivity of the hindpaw may increase due to a recently described phenomenon (*Tansley et al., 2019*) that we are calling hyperalgesic descending control of nociception (DCN), which represents the opposite of classical analgesic DCN (previously known as diffuse noxious inhibitory controls [DNIC] or conditioned pain modulation [CPM]). We use this term to describe the process by which pain in one region of an animal's body *increases* pain in another anatomically distinct region via descending pain modulatory circuitry (*Bannister et al., 2021*). Hindpaw sensitivity was measured in both contexts on days 1 and 4 of the experiment, 45 min following acetic acid injection. A 2 (Sex: male, female) × 2 (Context: vehicle-paired, acetic acid-paired) × 2 (Day: 1, 4) ANOVA revealed a main effect of day, $F_{(1, 14)} = 19.08$, MSE = 1.71, p = 0.001, $\eta p^2 = 0.58$, a day by sex interaction, $F_{(1, 14)} = 15.51$, MSE = 1.71, p = 0.009, $\eta p2 = 0.39$, a context by day interaction, $F_{(1, 14)} = 13.75$, MSE = 1.81, p = 0.002, $\eta p^2 = 0.50$, and a three-way interaction, $F_{(1, 14)} = 23.84$, MSE = 1.81, p < 0.001, $\eta p^2 = 0.63$. There was no main effect of context, sex, nor an interaction between the two (*F*s < 1). Planned comparisons were further conducted to assess differences both between and within-groups. Male mice (*Figure 1B*) exhibited similar levels of hindpaw mechanical sensitivity in both the acid- and vehicle-paired contexts on day 1 (p = 0.91) and day 4 (p = 0.29), suggesting that associative learning did not enable environmental control of pain sensitivity in males. Alternatively, hindpaw sensitivity of female mice varied greatly between chambers on days 1 and 4 of the paradigm (*Figure 1C*). On day 1, female mice exhibited hyperalgesic DCN (i.e., increased pain in the hindpaw following administration of a painful stimulus in the abdomen) in the acid-paired context relative to the vehicle chamber (p < 0.001). However, on day 4, female hindpaw sensitivity was greater in the vehicle-paired context than the acid-paired context (p = 0.001) despite the fact that animals received IP injection of 0.9% acetic acid prior to being placed into either chamber. The individual sexes exhibited similar levels of hindpaw sensitivity in the vehicle-paired chamber on day 1 (p = 0.72) but displayed both quantitative and qualitative differences in hindpaw sensitivity following the various acetic acid injections. Following acid injection on day 1, females exhibited hyperalgesic DCN whereas males did not (p < 0.001; *Figure 1D*). Similarly, hindpaw sensitivity in female mice was greater than in male mice following an IP injection of 0.9% acetic acid and placement into the vehicle-paired chamber on day 4 (p = 0.045), even though both sexes exhibited higher mechanical sensitivity in this chamber on day 4 as compared to day 1 (males: p = 0.003; females: p < 0.001). However, when injected with acid and placed into the acid-paired context on day 4, females exhibited less hindpaw sensitivity than males (p = 0.011; *Figure 1D*); male hindpaw withdrawal thresholds were lower in the acid-paired chamber on day 4 as compared to day 1 (p = 0.002), whereas female hindpaw withdrawal thresholds were higher in the acid-paired context on day 4 relative to day 1 (p = 0.006). This latter effect is especially interesting considering the dose of acetic acid administered in the acid-paired context on day 4 is ninefold higher than the dose administered in this same context on day 1.

In addition to hindpaw sensitivity, we also examined reflexive writhing behavior using the same 2 (Sex: male, female) × 2 (Context: vehicle-paired, acetic acid-paired) × 2 (Day: 1, 4) ANOVA. This analysis revealed a main effect of day, $F_{(1, 6)} = 198.76$, MSE = 53.07, p < 0.001, $\eta p^2 = 0.63$, but no other main effects or interactions (largest *F* = 1.00, p = 0.36). While no between- or within-subjects differences were observed on day 1 or day 4 in males (*Figure 1E*) or females (*Figure 1F*), males and females exhibited more writhing behaviors in both contexts on day 4, relative to day 1 likely because of the high dose of acetic acid administered prior to placement into either chamber on day 4 (*Figure 1G*;

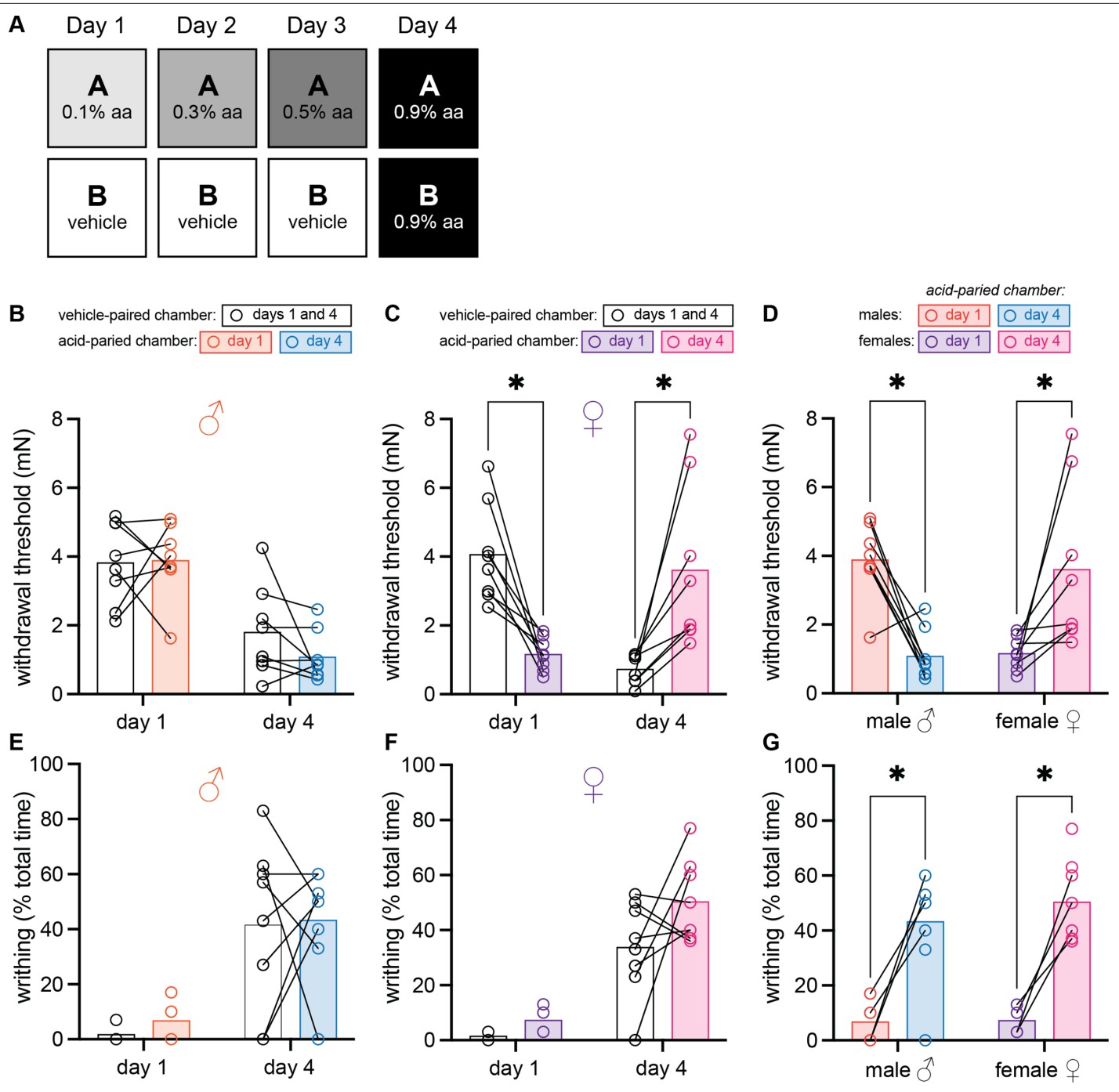

**Figure 1.** Female mice develop context-dependent analgesia after training with ascending doses of acetic acid. (**A**) Experimental design depicting the within-subject procedure. In daily sessions, mice were given ascending doses of acetic acid in one physical chamber, Context A, and vehicle injections in a separate physical chamber, Context B. On the final day, mice received an injection of 0.9% acetic acid solution in each context. Writhing behaviors were assessed on days 1 and 4 for the first 30 min following injection. Hindpaw sensitivity was measured 45 min following injection on days 1 and 4. (**B**) von Frey withdrawal thresholds of male mice (n = 8) on days 1 and 4 of the paradigm. Acetic acid injection on day 1 (0.1%) had no effect on hindpaw mechanical sensitivity. Hindpaw mechanical sensitivity was similar in both contexts following 0.9% acetic acid injection on day 4. (**C**) von Frey withdrawal thresholds of female mice (n = 8) on days 1 and 4 of the paradigm. Acetic acid injection on day 1 (0.1%) induced hindpaw mechanical hypersensitivity ('hyperalgesic descending control of nociception [DCN]'). Hindpaw mechanical sensitivity differed between the contexts following 0.9% acetic acid injection on day 4, however; females exhibited contextually mediated analgesia in the acid-paired chamber relative to the vehicle-paired chamber on day 4. (**D**) von Frey withdrawal thresholds of male and female mice replotted to highlight results in the acetic acid paired chamber. Male mice exhibit hyperalgesic DCN on day 4 relative to day 1. Female mice exhibit less hindpaw sensitivity following 0.9% acetic acid injection on day 4 than they did

*Figure 1 continued on next page*

*Figure 1 continued*

following 0.1% acetic acid injection on day 1 suggesting the development of conditioned analgesia. (**E**) Writhing behavior of male mice on days 1 and 4 of the paradigm. Writhing frequency was similar in both contexts on days 1 and 4. (**F**) Writhing behavior of female mice on days 1 and 4 of the paradigm. Writhing frequency was similar in both contexts on days 1 and 4. (**G**) Writhing behavior of male and female mice replotted to highlight results in acetic acid paired chamber. Both sexes exhibited more writhing behaviors following a 0.9% acid injection on day 4 than following a 0.1% acid injection on day 1.

male vehicle-paired 1 vs. 4: p = 0.005; male AA-paired 1 vs. 4: p = 0.014; female vehicle-paired 1 vs. 4: p = 0.03; female AA-paired 1 vs. 4: p = 0.003). Notably, context did not alter reflexive writhing behaviors in either sex on day 4 (*Figure 1G*), despite blocking the hyperalgesic DCN observed in von Frey testing. Collectively, these data suggest that prior experience, or pain memories associated with a specific environment, induce a conditioned compensatory response that decreases hyperalgesic DCN only in female mice.

## Female and male mice develop context-dependent analgesia after training with high doses of acetic acid

These initial experiments raised the possibility that males were not able to develop conditioned analgesia. However, unlike females, male mice also failed to develop mechanical hypersensitivity (i.e., hyperalgesic DCN) after injection of 0.1% acetic acid on day 1 of training (*Figure 1B*). Previous studies have also shown that male mice do not develop conditioned place aversion unless trained with doses of acetic acid >0.3% (*Bagdas et al., 2016*). Therefore, we repeated this within-subject paradigm but administered 0.9% acetic acid on all training and test days (*Figure 2A*) to determine if the UCS used in the previous experiments was simply not strong enough to support associative learning in males.

The same 2 (Sex: male, female) × 2 (Context: vehicle-paired, acetic acid-paired) × 2 (Day: 1, 4) ANOVA conducted to assess mechanical sensitivity found a marginal effect of day, $F_{(1, 14)}$ = 4.12, MSE = 1.54, p = 0.06, $\eta p2$ = 0.23, a day by sex interaction, $F_{(1, 14)}$ = 12.26, MSE = 1.54, p = 0.004, $\eta p^2$ = 0.47, and a context by day interaction, $F_{(1, 14)}$ = 40.63, MSE = 3.31, p < 0.001, $\eta p^2$ = 0.74. No other main effects or interactions were significant (largest $F$ = 1.95, p = 0.12). As before, planned comparisons were conducted to assess differences both between- and within-groups. In contrast to the results from the previous experiment, males exhibited hyperalgesic DCN in the acid-paired chamber on day 1 (p = 0.006; *Figure 2B*) and conditioned analgesia in this chamber on day 4 (p = 0.006) relative to the vehicle-paired chamber. A similar pattern was observed in females (*Figure 2C*); relative to the vehicle-paired chamber, female mice exhibited hyperalgesic DCN in the acid-paired chamber on day 1 (p = 0.003), and conditioned analgesia in this chamber as early as day 2 in some animals (*Figure 2—figure supplement 1*), but by day 4 in all mice (*Figure 2C*; p < 0.001). Although the magnitude of hyperalgesic DCN exhibited following 0.9% acetic acid injection on day 1 was greater in females than males (p = 0.024; *Figure 2D*), hindpaw mechanical thresholds in the acid-paired chamber on day 4 were not significantly different between the sexes (p = 0.078). In planned comparisons that assessed behavior across days, males exhibited hyperalgesic DCN in the vehicle-paired context on day 4 relative to day 1 (p < 0.001) but did not exhibit conditioned analgesia in the acid-paired chamber on day 4 relative to day 1 (*Figure 2D*; p = 0.42). This was not the case for females as they demonstrated both hyperalgesic DCN in the vehicle-paired context on day 4 relative to day 1 (p = 0.002) and conditioned analgesia in the acid-paired context on day 4 relative to day 1 (*Figure 2D*; p < 0.001).

Writhing behavior was also assessed using a 2 (Sex: male, female) × 2 (Context: vehicle-paired, acetic acid-paired) × 2 (Day: 1, 4) ANOVA. Main effects of context, $F_{(1, 10)}$ = 75.47, MSE = 95.24, p < 0.001, $\eta p^2$ = 0.88, and a context by day interaction were noted, $F_{(1, 10)}$ = 39.42, MSE = 292.09, p < 0.001, $\eta p^2$ = 0.80, but no other main effects or interactions were observed (largest $F$ = 3.13, p = 0.11). Planned comparisons revealed that males exhibited more writhing behavior in the acid-paired context than the vehicle-paired context on day 1 (*Figure 2E*, p < 0.001) but not day 4 (p = 0.56). Although females exhibited more writhing in the acid-paired context relative to the vehicle-associated context on day 1 (*Figure 2F*; p < 0.001), the opposite was observed on day 4 (p = 0.03); females exhibited less writhing in the acid-associated context than the vehicle-associated context despite receiving identical 0.9% acetic acid injections before being placed into each. Unsurprisingly, both males and females exhibited more writhing behaviors in the vehicle-paired context on day 4 relative to day 1 since animals were injected with 0.9% acetic acid before being placed into that chamber on

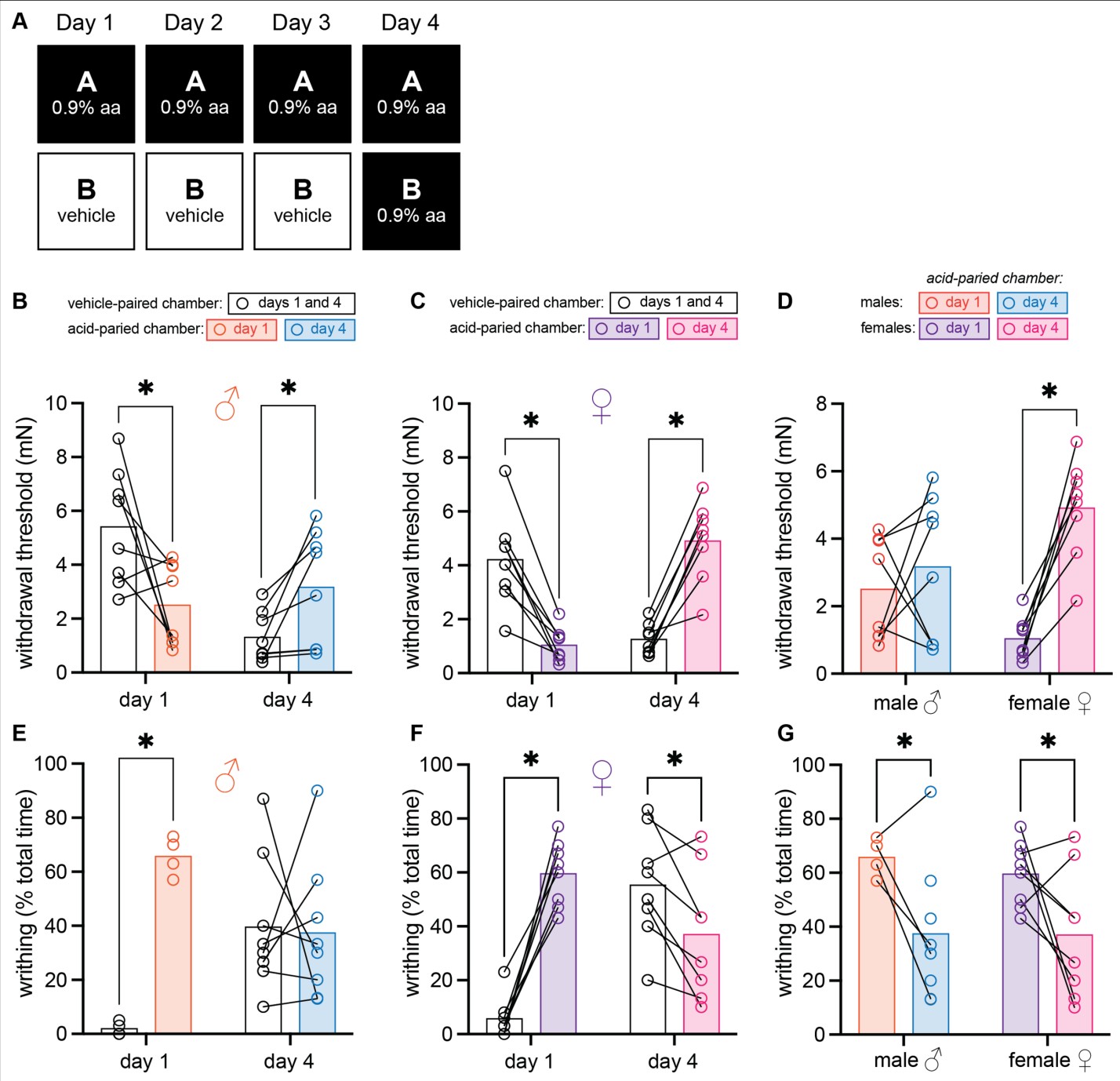

**Figure 2.** Female and male mice develop context-dependent analgesia after training with high doses of acetic acid. (**A**) Experimental design depicting the within-subject procedure. In daily sessions, mice were given a 0.9% dose of acetic acid in one physical chamber, Context A, and vehicle injections in a separate physical chamber, Context B. On the final day, mice received an injection of 0.9% acetic acid solution in each context. Writhing behaviors were assessed on days 1 and 4 for the first 30 min following injection. Hindpaw sensitivity was measured 45 min following injection. (**B**) von Frey withdrawal thresholds of male mice (n = 8) on days 1 and 4 of the paradigm. Acetic acid injection on day 1 (0.9%) induced hindpaw mechanical hypersensitivity ('hyperalgesic descending control of nociception [DCN]'). Hindpaw mechanical sensitivity differed between the contexts following 0.9% acetic acid injection on day 4, however; males exhibited contextually mediated analgesia in the acid-paired chamber relative to the vehicle-paired chamber on day 4. (**C**) von Frey withdrawal thresholds of female mice (n = 8) on days 1 and 4 of the paradigm. Acetic acid injection on day 1 (0.9%) induced hindpaw mechanical hypersensitivity ('hyperalgesic DCN'). Hindpaw mechanical sensitivity differed between the contexts following 0.9% acetic acid injection on day 4, however; females exhibited contextually mediated analgesia in the acid-paired chamber relative to the vehicle-paired chamber on day 4. (**D**) von Frey withdrawal thresholds of male and female mice replotted to highlight results in the acetic acid paired chamber. Male mice exhibited similar levels of hindpaw mechanical sensitivity in the acid-paired chamber on days 1 and 4. Female mice exhibited less hindpaw sensitivity

*Figure 2 continued on next page*

*Figure 2 continued*

following 0.9% acetic acid injection on day 4 than they did following 0.9% acetic acid injection on day 1 suggesting the development of conditioned analgesia. (**E**) Writhing behavior of male mice on days 1 and 4 of the paradigm. On day 1, injection of 0.9% acid induced more writhing than injection of vehicle. Writhing frequency of male mice was similar in both contexts on day 4. (**F**) Writhing behavior of female mice on days 1 and 4 of the paradigm. On day 1, injection of 0.9% acid induced more writhing than injection of vehicle. Writhing frequency of female mice was lower in the acid-paired context on day 4 than in the vehicle-paired context. (**G**) Writhing behavior of male and female mice replotted to highlight results in acetic acid paired chamber. Both sexes exhibited less writhing on day 4 relative to day 1.

The online version of this article includes the following figure supplement(s) for figure 2:

**Figure supplement 1.** Time course of conditioned analgesia development in male and female mice.

day 4 (males: p = 0.02; females: p < 0.001). However, both sexes exhibited less writhing in the acid-paired context on day 4 relative to day 1 suggesting either the development of acetic acid tolerance or conditioned analgesia (*Figure 2G*; males: p = 0.043; females: p = 0.047). Considering these data, and the fact that male mice had higher withdrawal thresholds in the acid-paired context on day 4 as compared to the vehicle-paired context, we conclude that both sexes are able to develop conditioned analgesia if trained with strong UCS, but the magnitude of this phenomenon may differ between the sexes with females being especially sensitive to this type of learning.

## Conditioned analgesia is unlikely to be mediated by changes in circulating corticosterone, but rather increased endogenous opioid signaling

To begin probing the biological basis of this conditioned analgesia, we next developed a between-subject 3-day conditioning paradigm in which animals were only trained in one environment so as to isolate physiological changes occurring in each context (*Figure 3A*). Since male and female mice both exhibited conditioned analgesia in the previous experiment, sexes were combined for this between-subject analysis. A 2 (Group: acid-trained, vehicle-trained) × 2 (Day: 1, 4) ANOVA found a day by group interaction, $F_{(1, 14)} = 49.30$, MSE = 2.01, p < 0.001, $\eta p^2 = 0.78$, but no main effect of group or day (largest $F = 1.43$, p = 0.25). Planned comparisons found that similar to observations in the acid-paired context in our within-subject experiment (*Figure 2*), acid-trained animals exhibited hyperalgesic DCN relative to vehicle-trained animals on day 1 (p < 0.001) and conditioned analgesia relative to vehicle-trained animals on day 4 (p = 0.008; *Figure 3B*). To determine if this context-dependent analgesia is associated with stress, circulating corticosterone was measured in animals immediately upon removal from the testing environment. Corticosterone levels were similar between acid- and vehicle-trained mice (*Figure 3C*; $t_{(14)} = 0.56$, p = 0.58) suggesting that the observed behaviors were likely not forms of stress-induced analgesia or hyperalgesia in the acid- and vehicle-trained mice, respectively.

Another possible biological explanation for conditioned analgesia is that after repeated exposure to acetic acid, the context may promote the release of endogenous opioids as part of a compensatory response to the upcoming acetic acid treatment. To test this hypothesis, we repeated the between-subject 3-day conditioning paradigm, but injected animals with naloxone, a broad-spectrum opioid receptor antagonist, or vehicle before they were placed into the testing environment on day 4 (*Figure 3D*). A 2 (Group: vehicle-trained, acid-trained) × 2 (Drug: vehicle, naloxone) ANOVA found a main effect of group, $F_{(1, 28)} = 6.68$, MSE = 2.77, p = 0.015, $\eta p^2 = 0.19$, a main effect of drug, $F_{(1, 28)} = 6.67$, MSE = 2.77, p = 0.015, $\eta p^2 = 0.19$, and an interaction between the two, $F_{(1, 28)} = 11.57$, MSE = 2.77, p = 0.002, $\eta p^2 = 0.29$. We found no evidence for any effect of naloxone pre-treatment on the hyperalgesic DCN exhibited by vehicle-trained animals on day 4 (*Figure 3E*; p = 0.64), but it successfully blocked the conditioned analgesia exhibited by acid-trained mice on day 4 (*Figure 3E*; p < 0.001); the mechanical withdrawal thresholds of naloxone injected acid-trained mice were similar to those exhibited by vehicle-trained mice after their first exposure to acetic acid. To determine if naloxone treatment blocked conditioned analgesia to a similar extent in acid-trained males and females, we performed a 2 (Sex: male, female) × 2 (Drug: vehicle, naloxone) ANOVA. This analysis revealed a main effect of drug, $F_{(1, 20)} = 31.62$, MSE = 3.14, p < 0.001, $\eta p^2 = 0.61$, but no main effect of sex nor an interaction between the two (*F*s < 1), suggesting that naloxone blocked conditioned analgesia to a similar extent in male (vehicle vs. naloxone; p = 0.005) and female (vehicle vs. naloxone; p < 0.001) acid-trained mice (*Figure 3F*).

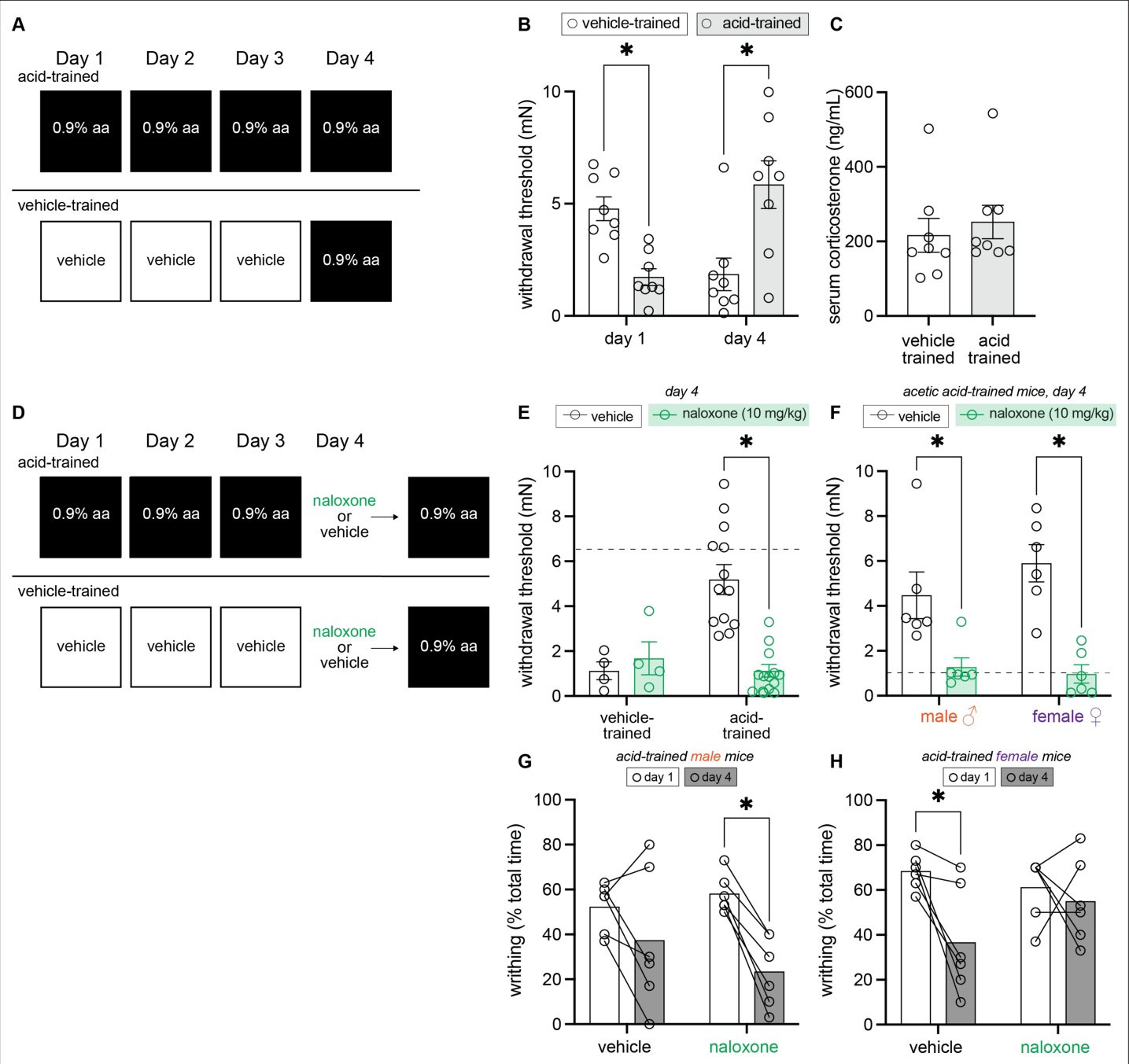

**Figure 3.** Endogenous opioid signaling, and not circulating corticosterone, is associated with contextually mediated conditioned analgesia. (**A**) Experimental design depicting the between-subject design. Animals received an intraperitoneal (IP) injection of either acetic acid or vehicle for 3 days. On the fourth day, all animals were given an IP injection of 0.9% acetic acid. (**B**) von Frey withdrawal thresholds of male and female mice on days 1 and 4 of the paradigm. Animals in the vehicle-trained group (n = 8; 4 males, 4 females) have higher withdrawal thresholds than those injected with acid on day 1. On day 4, vehicle-trained mice (n = 8; 4 males, 4 females) exhibit 'hyperalgesic descending control of nociception [DCN]' and have lower withdrawal thresholds than acid-trained mice. Acid-trained mice exhibit 'hyperalgesic DCN' on day 1 and conditioned analgesia on day 4. (**C**) Results from ELISA corticosterone assay. Groups (n = 8; same mice from panel B) did not differ in their circulating cortisol levels 60 min following acid injection. (D) Experimental design depicting the between-subjects design. Animals received an IP injection of either acetic acid or vehicle for 3 days. On the fourth and final day, all animals were given an IP injection of acetic acid. However, half of the animals in each condition were pre-treated with naloxone, an opioid receptor antagonist, while the other half received vehicle. (**E**) von Frey withdrawal thresholds of male and female mice on day 4. Animals in the vehicle-trained groups (n = 4) showed mechanical hypersensitivity irrespective of naloxone pre-treatment, and animals in the acid-trained group (n = 12) exhibited conditioned analgesia that was blocked by naloxone. The dashed gray line represents the average withdrawal threshold of vehicle-

*Figure 3 continued on next page*

*Figure 3 continued*

trained mice on day 1. (**F**) von Frey withdrawal thresholds of acid-trained male (n = 6 per treatment group) and female (n = 6 per treatment group) mice replotted to examine sex differences on day 4. Both males and females showed increased mechanical hypersensitivity when pre-treated with naloxone, suggesting that contextually mediated analgesia is mediated by the endogenous opioid system. The dashed gray line indicates the average withdrawal threshold of vehicle-trained mice on day 4. (**G**) Writhing frequency of acid-trained male mice on days 1 and 4 of the paradigm. On day 4, naloxone pre-treated mice exhibited a decrease in writhing as compared to day 1. (**H**) Writhing frequency of acid-trained female mice on days 1 and 4 of the paradigm. On day 4, vehicle pre-treated mice exhibited a reduction in writhing frequency as compared to day 1. Naloxone pre-treatment prevented this decrease in writhing on day 4, suggesting the reduction observed in vehicle-treated animals is a form of conditioned analgesia.

A 2 (Sex: male, female) × 2 (Drug: vehicle, naloxone) × 2 (Day: 1, 4) ANOVA conducted to analyze writhing behaviors, however, revealed divergent effects of naloxone in each sex; a main effect of day, $F_{(1, 20)} = 22.60$, MSE = 255.08, p < 0.001, $\eta p^2 = 0.53$, a main effect of sex, $F_{(1, 20)} = 4.69$, MSE = 400.14, p = 0.043, $\eta p^2 = 0.19$, and a three-way interaction, $F_{(1, 20)} = 6.04$, MSE = 255.08, p = 0.023, $\eta p^2 = 0.23$ were observed. Similar to results in the within-subject experiment, vehicle-treated acid-trained male mice showed a modest reduction in writhing from day 1 to day 4 (*Figure 3G*; p = 0.12), hinting at either the development of acid tolerance or conditioned analgesia. This reduction in writhing behavior was even more pronounced in naloxone-treated acid-trained animals, suggesting that decreases in reflexive writhing behaviors are not a result of increased endogenous opioid signaling in males (*Figure 3G*; p = 0.001). Statistically, no difference in writhing behaviors was noted between naloxone- and vehicle-treated, acid-trained male mice on day 4 (p = 0.31), thus supporting the idea that unlike hyperalgesic DCN, reflexive visceral pain behaviors in male mice may not be modulated by endogenous opioid signaling. Alternatively, vehicle-treated, acid-trained females exhibited conditioned analgesia on day 4 relative to day 1 (*Figure 3H*; p = 0.003), an effect that was blocked by naloxone treatment on day 4 (*Figure 3E*; p = 0.51). Collectively, these results suggest that both male and female mice recruit the endogenous opioid system as a compensatory response to repeated acetic acid administration, however the extent to which this system influences UCS-induced pain behaviors may differ between the sexes.

## Contextual cues previously associated with pain induce conditioned hypersensitivity in male mice and conditioned analgesia in female mice

Up to this point, all experimental paradigms included acid injection (UCS) on the final testing day. As a result, it was unclear if the compensatory analgesic response was truly context-dependent (i.e., a CR) or simply an autonomic response initiated by repeated acetic acid treatment. To answer this question, we developed a within-subject, 3-day training paradigm that concluded with vehicle administration before placement in either training context on day 4 (*Figure 4A*). Withdrawal thresholds were analyzed using a 2 (Sex: male, female) × 2 (Context: vehicle-paired, acetic acid-paired) × 2 (Day: 1, 4) ANOVA. This analysis found a main effect of context, $F_{(1, 14)} = 46.45$, MSE = 1.10, p < 0.001, $\eta p^2 = 0.77$, a day by sex interaction, $F_{(1, 14)} = 4.79$, MSE = 4.30, p = 0.046, $\eta p^2 = 0.26$, and a context by day interaction, $F_{(1, 14)} = 5.65$, MSE = 4.78, p = 0.032, $\eta p^2 = 0.29$, but no other main effects or interactions (largest F = 2.89, p = 0.11).

Male mice exhibited hyperalgesic DCN in the acid-paired context relative to the vehicle-paired context on day 1 (*Figure 4B*; p < 0.001), and, similar to the conditioned hypersensitivity previously reported after single exposure to a noxious UCS (*Martin et al., 2019*), context-dependent hypersensitivity in the absence of the UCS on day 4 (*Figure 4B*; p = 0.027). Male hindpaw withdrawal thresholds in the acid-paired chamber were lower than thresholds recorded in the vehicle-paired chamber on day 4, despite receiving vehicle injections before being placed into either chamber. This conditioned hypersensitivity was not observed in females. Despite exhibiting the same hyperalgesic DCN as males on day 1 (*Figure 4C*; p < 0.001), female mice exhibited the same level of mechanical sensitivity in the vehicle- and acid-paired contexts on day 4 (*Figure 4C*, p = 0.81). Further, in contrast to male mice that exhibited similar withdrawal thresholds in the acid-paired chamber on days 1 and 4, female hindpaw withdrawal thresholds in the acid-paired chamber were higher on day 4 than on day 1 (p = 0.003), suggesting either a lack of any CR, or the development of conditioned analgesia.

To determine if female behavior on day 4 was reflective of context-dependent endogenous opioid release, and thus a form of conditioned analgesia, we performed a between-subject, 3-day training paradigm with either acetic acid or PBS vehicle and then treated animals with naloxone prior to being

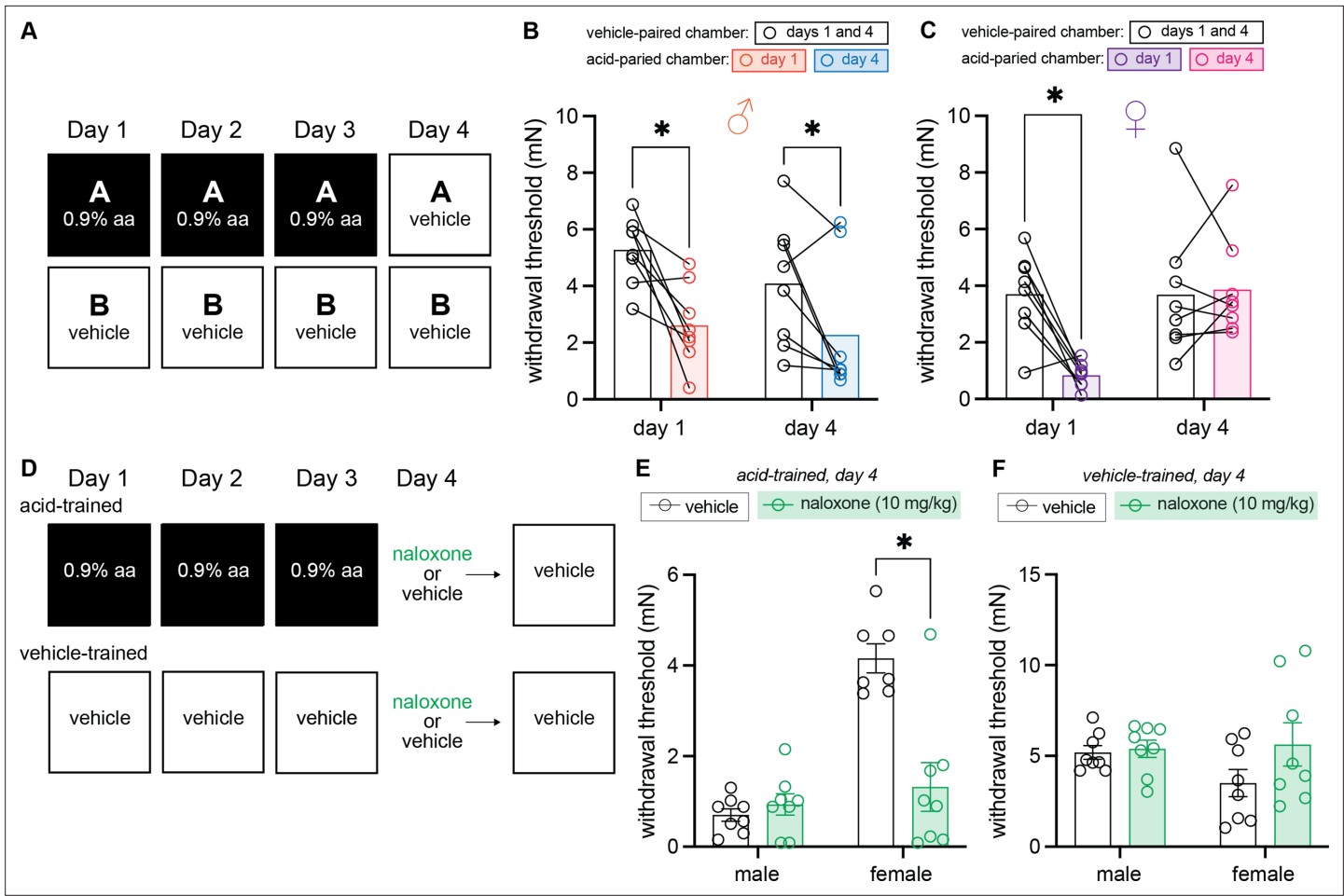

**Figure 4.** Males exhibit conditioned hypersensitivity and females exhibit conditioned analgesia in the absence of unconditioned stimulus (UCS) administration. (**A**) Experimental design depicting the within-subject procedure. In daily sessions, mice were given a 0.9% dose of acetic acid in one physical chamber, Context A, and vehicle injections in a separate physical chamber, Context B. On the final day, mice received an injection of vehicle in each context. Hindpaw sensitivity was measured 45 min following injection on days 1 and 4. (**B**) von Frey withdrawal thresholds of male mice (n = 8) on days 1 and 4 of the paradigm. Acetic acid injection on day 1 (0.9%) induced hindpaw mechanical hypersensitivity ('hyperalgesic descending control of nociception [DCN]'). Similar hindpaw mechanical hypersensitivity was observed in the acid-paired chamber on day 4, despite the fact that animals received a vehicle injection prior to being placed in this chamber. (**C**) von Frey withdrawal thresholds of female mice (n = 8) on days 1 and 4 of the paradigm. Acetic acid injection on day 1 (0.9%) induced hindpaw mechanical hypersensitivity ('hyperalgesic DCN'). Hindpaw mechanical sensitivity did not differ between the contexts following vehicle injection on day 4. (**D**) Experimental design depicting the between-subject procedure. In daily sessions, mice were given a 0.9% dose of acetic acid or vehicle injection prior to being placed into one physical chamber. On the final day, mice received an injection of naloxone or vehicle prior to being placed into the same training context; no acid injection was performed on day 4. (**E**) von Frey withdrawal thresholds of acid-trained male (n = 8) and female (n = 7–8) mice on day 4 of the paradigm. Naloxone pre-treatment did not affect the hindpaw withdrawal thresholds of acid-trained male mice. Conversely, naloxone treatment decreased the hindpaw withdrawal thresholds of acid-trained female mice, suggesting conditioned recruitment of endogenous opioid systems in the training context. (**F**) Withdrawal thresholds of vehicle-trained male (n = 8) and female (n = 8) mice on day 4 of the paradigm. Naloxone pre-treatment had no effect on hindpaw sensitivity of male or female mice after 3 days of context training.

placed in the testing context on day 4 (**Figure 4D**). Mechanical sensitivity was assessed on day 4 in acid-trained animals using a 2 (Sex: male, female) × 2 (Drug: vehicle, naloxone) ANOVA. This found main effects of both sex, $F_{(1,27)} = 31.42$, MSE = 0.91, $p < 0.001$, $\eta p^2 = 0.54$ and drug, $F_{(1,27)} = 14.44$, MSE = 0.91, $p < 0.001$, $\eta p^2 = 0.35$, and an interaction between the two, $F_{(1,27)} = 20.07$, MSE = 0.91, $p < 0.001$, $\eta p^2 = 0.43$. Naloxone treatment had no effect on the conditioned hypersensitivity exhibited by male mice on day 4 (**Figure 4E**; $p = 0.63$), further supporting the notion that this behavior is an example of stress-induced hyperalgesia. When administered in females however, naloxone unmasked an endogenous opioid-mediated conditioned compensatory response (**Figure 4E**; $p < 0.001$). In the absence of naloxone treatment, female behaviors appear to be unaffected by 3 days of acid training

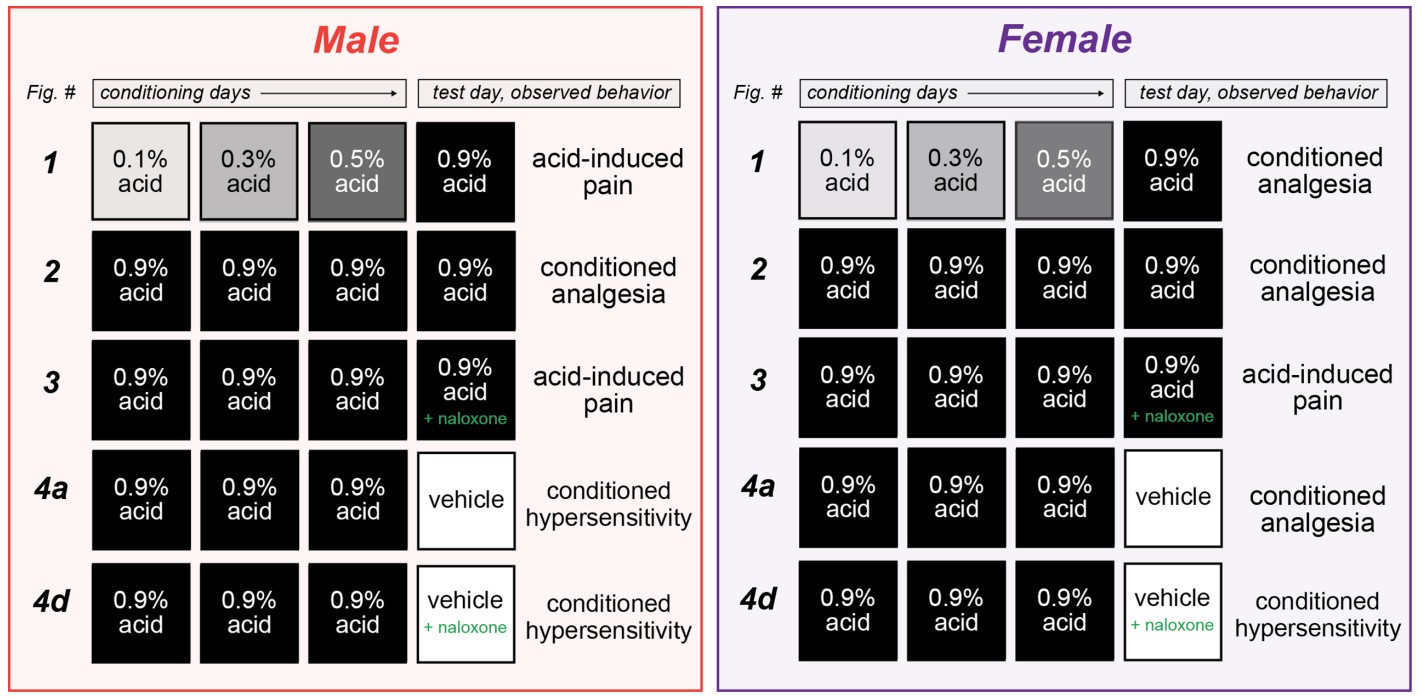

**Figure 5.** Summary of sexually divergent conditioned pain behaviors. Graphical depiction of conditioning trials, test day manipulations, and subsequent behavior results. After training with high doses of acetic acid, male mice exhibited opioid-dependent conditioned analgesia when presented with the same noxious unconditioned stimulus (UCS) (row 2), and conditioned hypersensitivity when placed into the context where conditioning occurred (row 4a). Alternatively, female mice developed opioid-dependent conditioned analgesia after training with low (row 1) or high (row 2) doses of acetic acid. This conditioned analgesia was also observed when animals were placed into the context where conditioning occurred, even in the absence of the UCS (row 4a).

(i.e., females had high withdrawal thresholds in the acid-trained context on day 4 relative to day 1 as they did not receive an acetic acid injection prior to entering the context on day 4; *Figure 4C*). However, in the absence of acetic acid injection on day 4, naloxone treatment decreased withdrawal thresholds suggesting that females actually exhibit context-dependent conditioned hypersensitivity that is readily counteracted with activation of their endogenous opioid system. To determine if contextual recruitment of this system occurs in the absence of noxious UCS training, a 2 (Drug: vehicle, naloxone) × 2 (Sex: male, female) ANOVA was completed on results recorded in vehicle-trained animals (*Figure 4F*). No main effect of sex, $F < 1$, drug, $F_{(1, 28)} = 2.30$, p = 0.14, or an interaction between the two, $F_{(1, 28)} = 1.59$, p = 0.22 was observed, suggesting that endogenous opioid systems are not engaged by the environment unless that environment is coupled with a painful experience.

## Discussion

In the present set of experiments (see *Figure 5* for a summary), we asked whether environment can exert control over pain sensitivity and pain expectation through associative learning. We first demonstrated, in a within-subject manner, that a context (CS) paired with a painful experience (acetic acid injection; UCS) can induce conditioned analgesia in animals upon re-exposure to the CS. Although female mice demonstrate this conditioned analgesia after pairing a relatively weak UCS with the CS, male mice only develop conditioned analgesia if a stronger UCS is used throughout conditioning trials. We then determined that this effect is likely not mediated by stress, but rather by activation of the endogenous opioid system since administration of the opioid antagonist naloxone blocked conditioned analgesia in both sexes. In a second set of experiments, we demonstrated that males exhibit conditioned hypersensitivity when placed into a context previously associated with pain. Alternatively, females exhibit conditioned analgesia that is dependent on endogenous opioid signaling when placed into a similar environment. Together, these results demonstrate that context can come to exert

control over the endogenous opioid system such that it is recruited as a conditioned compensatory response to ongoing painful stimuli in both sexes, or in anticipation of a painful stimulus in females.

In the ongoing search for better/novel analgesics, many preclinical biologists, including some of the present authors, have taken a reductionist approach using highly standardized in vitro or in vivo assays to determine how particular genes or proteins contribute to the process of nociception. In so doing, significantly less has been learned about the influence external factors have on pain perception and the development of chronic pain. In their 'Imprecision Hypothesis' (2015), Moseley and Vlaeyen posit that chronic pain may develop, in part, as a result of associative learning; repeated coupling of a UCS – in this case a noxious input – with neutral cues like a specific movement of the body (e.g., bending over) or context may lead to the perception of pain (CR) upon subsequent presentation of these newly CS. Data presented in this manuscript are, to our knowledge, the first preclinical test of this hypothesis; we repeatedly paired noxious visceral stimulation (i.e., IP acetic acid injection; UCS) with a unique context (i.e., a behavior chamber; CS). Similar to the results obtained following a single pairing of this same UCS and CS (*Martin et al., 2019*), male mice developed conditioned hypersensitivity upon CS presentation alone (*Figure 4B*). In contrast, when females are exposed to the CS, they engage their endogenous opioid system, such that behavioral responses appear similar to those exhibited prior to injury or conditioning (*Figure 4C*). These divergent results suggest there are qualitative sex differences in the biological processes that mediate pain expectation.

The biological basis and evolutionary explanation for these qualitative sex differences is currently unknown. Dimorphisms in endogenous opioid circuit anatomy and function are well documented (*Chakrabarti et al., 2010*; *Liu et al., 2007*; *Loyd and Murphy, 2006*; *Loyd et al., 2008*; *Tershner et al., 2000*). Perhaps adaptive pressures associated with the reproduction timeframe and energy requirement led to differential abilities to recruit endogenous opioid signaling such that female organisms are better able to activate this system as a means of dampening ongoing pain. Another intriguing hypothesis is that brain regions required for contextual conditioning – like the dorsal hippocampus (*Maren and Holt, 2004*; *Matus-Amat et al., 2004*) and retrosplenial cortex (*Keene and Bucci, 2008*; *Kwapis et al., 2015*; *Miller et al., 2021*; *Trask et al., 2021*) – and those involved in descending pain inhibition are differentially connected in each sex. For example, electrical stimulation of the retrosplenial cortex decreases both formalin-associated spontaneous pain (*Reis et al., 2010*) and postsurgical mechanical hypersensitivity (*Rossaneis et al., 2011*) by modulating opioid signaling in the anterior pretectal nucleus (*Rossaneis et al., 2011*). Because only male rats were used in these studies, it is unclear (1) if this circuit functions similarly in female rodents, (2) if this circuit is engaged during contextual conditioning, and (3) if the retrosplenial cortex, or other structures important for contextual learning, differentially projects to other brain regions associated with endogenous analgesia modulation in the two sexes.

In addition to assessing pain expectation, we also examined if associative learning impacts pain sensitivity, or the magnitude of the UCR elicited by a noxious stimulus. The UCS used in these experiments was IP injection of acetic acid, a procedure that induces reflexive writhing behaviors (*Koster et al., 1959*). In addition to writhing, hindpaw mechanical sensitivity testing was used as a second UCR to examine a behavioral output that, by definition in this experimental paradigm, is modulated by descending control. Recently, *Tansley et al., 2019*, characterized a phenomenon in which pain in one part of the body causes hyperalgesia in an anatomically separate body region if evoked with a relatively mild noxious stimulus. In contrast to the 'pain-inhibits-pain' idea associated with the electrophysiological phenomenon known as DNIC, this observation which has been called 'anti-DNIC', facilitatory CPM, or 'hyperalgesic DCN' is observed across multiple rodent species and mouse strains and does not depend on the anatomical proximity of the injured and test sites. Despite initially demonstrating hyperalgesic DCN, both male and female mice in the present experiments developed conditioned analgesia after repeated exposure to the noxious stimuli; both sexes exhibited less pain when presented with a noxious UCS in an environment (CS) associated with an acute pain memory (*Figure 2B and C*). Notably, however, there was a quantitative sex difference in the strength of the UCS required to develop this conditioned pain tolerance; male mice only developed conditioned pain tolerance if trained with high doses of acetic acid. Strength of conditioning also influenced whether conditioned analgesia was observed in writhing behaviors. Context (CS) had no effect on writhing behaviors except in female mice trained with high doses of acetic acid (*Figure 2F*). Conditioned analgesia was not observed in writhing data collected from female mice trained with lower doses of acetic

acid, or male mice trained with high or low doses of acid. One potential explanation for the observed differences in writing and hindpaw withdrawal behaviors is that writhing may reflect the reflexive unconditional response to acetic acid (analogous to rodent jumping in response to a shock presentation; animals habituate to stimuli with repeated presentations) whereas the sensitivity of the hindpaw might be a more direct assessment of pain expectancy (analogous to freezing to a tone previously paired with shock). Regardless, these data reiterate the idea that females are particularly sensitive to, and thus able to associate, external cues more strongly with visceral pain stimuli. This is perhaps unsurprising given the extensive literature describing sex differences in pain perception.

The idea that males and females perceive pain differently is a long-standing topic in both academic and popular science discussions. Upon review of this topic, it is clear that women are more sensitive to acute pain stimuli than men, and that chronic pain is more frequently reported in women than it is in men (*Mogil, 2012*; *Mogil, 2020*). Rodent data presented in this paper agree with this conclusion; female mice exhibited more pain than male mice following low (0.1%; *Figure 1D*) and high (0.9%; *Figure 2D*) acetic acid administration. Similar quantitative sex differences have been previously reported in other visceral pain models. For example, male and ovariectomized female rodents are less sensitive to chemical or mechanical injury of the colon and urinary bladder (reviewed by *Traub and Ji, 2013*). Combined with the fact that stronger UCS will support more conditioning to stimuli presented with those UCS (*Holland, 1979*; *Morris and Bouton, 2006*; see *Rescorla and Wagner, 1972*), the present results suggest that females more readily associate contextual cues with noxious visceral stimuli because they perceive visceral pain more intensely than males. Under this logic, females should develop conditioned analgesia more quickly than males if the same noxious stimulus is used to train both sexes. In support of this hypothesis, six of eight female mice included in *Figure 2* exhibited conditioned analgesia on day 2 of the paradigm (i.e., higher paw withdrawal thresholds in the acid-paired chamber on day 2 vs. day 1) whereas zero of the eight male mice in this same study exhibited conditioned analgesia on day 2 (*Figure 2—figure supplement 1*). An additional interpretation of the present results is that the visceral pain introduced via acetic acid injection created a secondary internal context that males are unable to generalize. A wide variety of internal states have been shown to serve as contextual stimuli that influence behavioral performance (*Garfinkel et al., 2021*; *Schepers and Bouton, 2017*; *Schepers and Bouton, 2019*; see *Bouton, 1991*). Under this interpretation, the present results, in which males show conditioned hypersensitivity when placed into the acid-paired context without the UCS, might be explained by a lack of generalization as only the exteroceptive, but not interoceptive, context is the same as that which was included in training. While this seems unlikely to explain all results, interoceptive contexts might have played a role in the current design.

Strength of conditioning, or lack of UCS generalization, may also have influenced whether conditioned hyperalgesia or analgesia was displayed in the training context in the absence of the UCS. For example, after one pairing of 0.9% acetic acid and context, male mice exhibit conditioned hypersensitivity (*Martin et al., 2019*) whereas females, based on the observations made in these studies, likely exhibit conditioned analgesia. Would females exhibit conditioned hypersensitivity if trained with a single exposure to 0.1% acetic acid? Would males exhibit conditioned analgesia if trained with more than three exposures to 0.9% acetic acid? Experiments of this nature would allow for the classification of these sex differences as qualitative or quantitative. To this end, it is also important to clarify how long conditioned analgesia/hyperalgesia last in the absence of further conditioning and to determine the stimulus intensity and frequency required to maintain conditioning. Answers to these questions might allow for better explanations as to why chronic pain is more prevalent in females than males and provide opportunities to develop pain therapies based on associative learning principles. Although the opportunity for maladaptive learning exists, females may be better prepared, biologically speaking, to associate relief of ongoing pain with CS than males. Confirmation of this hypothesis may lead to the development of novel cognitive therapeutic strategies for pain that rely on the principles of Pavlovian conditioning, or those related to other memory-based manipulations like reframing (*Pavlova et al., 2022*). However, these approaches should be developed with caution. In the present study, mice were exposed to only novel CS and UCS. It is a well-documented finding that previous exposure to a putative CS makes it more difficult to become predictive of a UCS (termed 'latent inhibition'; e.g., *Lubow, 1973*). In cases of chronic pain development in humans, it is rarely the case that either the environment or the chronic pain are inherently novel, and thus, this type of conditioning would likely take longer to develop than in the procedure employed here.

In conclusion, the data presented here demonstrate that context can control pain perception while an animal is actively experiencing pain or when an animal is anticipating a painful experience. These results are evidence that pain perception and engagement of endogenous opioid systems can become dependent on the environment for expression and can be modified through their psychological association with environmental cues. Future work will examine the neural pathways associated with the contextual control of pain processing including those that mediate opioid release in response to pain-associated environments. Further, additional studies will need to examine if continuous chronic pain is susceptible to the effects of conditioning observed here.

## Materials and methods

### Animals

Equally sized cohorts of male and female C57BL/6 mice aged 8–12 weeks were used in all experiments. Mice were either bred in house or purchased from The Jackson Laboratory (Bar Harbor, ME). Purchased animals acclimated to the housing facility for >7 days before experimental use. Purchased animals and animals bred in-house were not intermixed, but rather used in independent experiments. Animals were randomized to treatment group. All protocols were in accordance with National Institute of Health guidelines and were approved by the Institutional Animal Care and Use Committee at the Medical College of Wisconsin (Milwaukee, WI; protocol #0383). We are not the first to use repeated acetic acid injections as painful stimuli. This procedure was previously used in *Stevenson et al., 2006*; approximately 4 hr after acetic acid injection, animals no longer exhibited acetic acid-induced feeding suppression in those studies. Similarly, no animal in our study lost more than 15% of their body weight over the course of the experiments. Therefore, no animals were excluded from these studies as dictated by approved IACUC endpoints.

### Conditioning paradigms

All experiments used two distinct contexts, counterbalanced as Context A and B for all animals. The first context was a $10 \times 10 \times 15$ cm$^3$ Plexiglas chamber. Two walls of the chamber were solid black Plexiglas, and a horizontal white and black striped pattern covered the remaining two walls. These chambers were cleaned with C-DOX, a chlorine-based disinfectant. The second chamber was a similarly sized Plexiglas chamber, consisting of four black walls, and cleaned with 70% ethanol. Both contexts were placed onto a raised 0.7 cm$^2$ wire platform to allow for video recordings and mechanical sensitivity testing. Dilute acetic acid (0.1%, 0.3%, 0.5%, or 0.9%) was used as the UCS in all experiments. Animals received an IP injection (10 mL/kg body weight) of acetic acid (purchased from Sigma Aldrich; diluted in phosphate buffered saline, PBS, pH 7.4) or PBS vehicle according to the experimental details described below. The 0.9% dose of acetic acid was used as the final UCS since it has previously been shown to elicit conditioned pain hypersensitivity in male mice (*Martin et al., 2019*).

#### Within-subject assessment of conditioned pain tolerance

On days 1, 2, and 3, animals received an IP injection of acetic acid immediately before being placed into one of the two contexts for a total of 60 min. In the escalating dose experiment (*Figure 1*), the concentrations administered were 0.1%, 0.3%, and 0.5% acetic acid on days 1, 2, and 3, respectively. For the consistent dose experiments (*Figure 2*), a 0.9% acetic acid injection was given on days 1, 2, and 3. Animals received an equivalent volume injection of PBS immediately before being placed into the second of the two contexts for a total of 60 min. Contexts and time of injection (i.e., morning or afternoon) were counterbalanced across all animals and context pairings were separated by 3 hr for each animal. On day 4, animals received an IP injection of 0.9% acetic acid immediately before being placed into Context A. A second IP injection of 0.9% acetic acid was administered 3 hr later immediately before placing the animal into Context B. Pain behavior testing was completed after being placed into either context as described below.

#### Between-subject assessment of conditioned pain tolerance

Between-subjects experiments were similar to the within-subject experiments, with the exception that each animal was only exposed to one context for 60 min each day. On days 1, 2, and 3, half of all animals received an IP injection of 0.9% acetic acid immediately before being placed into one of the

two conditioning chambers described above; chambers were counterbalanced between groups. The remaining half of the animals received an IP injection of PBS before being placed into the chamber. On day 4, all animals received an IP injection of either 0.9% acetic acid (*Figures 3E–H and 4E*) or PBS (*Figures 3E and 4F*) immediately before being placed into their training context. Pain behavior testing was completed after being placed into the context on days 1 and 4 as described below.

## Pain behavioral measures

Animals were moved to the behavior testing room at approximately 07:00 hr each morning. Overhead lights were on throughout the entirety of the habituation and testing period. Animals remained in their home cages in the behavior room for no less than 1 hr prior to the first injection each day. Immediately following acetic acid or vehicle injection, animals were placed into Context A or Context B. A video recording device placed beneath the animals captured all movements for 30 min following injection. Videos were scored offline by an experimenter blinded to treatment. Videos were sampled once every minute for 10 s for the presence or absence of the writhing posture as previously described (*Martin et al., 2019*). Writhing was reported for each animal as a percentage of total bins in which the posture was observed. Mechanical sensitivity was assessed in each animal 45–60 min following injection using von Frey filaments. Calibrated monofilaments were delivered through the wire testing platform and applied to the plantar surface of each hindpaw following the up-down method (*Dixon, 1965*); the 50% withdrawal threshold of each paw was calculated then averaged between paws (*Dixon, 1980*; *Chaplan et al., 1994*). Toe flaring was not considered a withdrawal. von Frey filaments were not applied to the hindpaw if animals were actively writhing.

## Naloxone treatment

Naloxone hydrochloride was purchased from Sigma Aldrich and dissolved in PBS, pH 7.4. Naloxone (10 mg/kg) or an equivalent volume of PBS vehicle was administered via IP injection 5 min before 0.9% acetic acid (*Figures 3E–H and 4E*) or vehicle (*Figures 3E and 4F*) was injected. Animals were returned to their home cages between the two injections. This dose (10 mg/kg) has previously been shown to block the contribution of endogenous opioid signaling in the tail-withdrawal test (*Rosen et al., 2019*) and reduce spontaneous pain behaviors initiated by complete Freund's adjuvant injection (*Lee et al., 2021*).

## Corticosterone ELISA

Immediately following von Frey mechanical sensitivity testing (i.e., 60 min following acetic acid or vehicle injection on day 4; *Figure 3A–B*), animals were removed from the testing apparatus, placed into a clean shoebox cage, and transferred (<1 min) to a neighboring necropsy room. Trunk blood was obtained from isoflurane anesthetized mice via cardiac puncture, transferred to a chilled tube containing 3.5% sodium citrate, then centrifuged at 1500 rpm, 4°C for 15 min. Plasma was collected and stored at –80°C until analysis. Plasma concentrations of corticosterone were measured using the Corticosterone Enzyme Immunoassay kit (Arbor Assay's DetectX) as previously described (*Long et al., 2016*).

## Data reporting and analysis

Data presented in this manuscript are those collected during the first performance of each experiment. Results for all animals enrolled in each experiment are reported; no outliers were encountered. All data were analyzed using repeated measures ANOVAs with α = 0.05 in SPSS v.28. Planned comparisons were performed following ANOVAs to determine between- and within-group differences if significant interactions or main effects were observed. Estimates of effect size were calculated using a partial eta-squared. Power analyses were used to determine appropriate sample sizes. Means (μ), standard deviations (σ), and expected effect sizes were obtained from preliminary studies and data of a similar nature previously published by the Stucky Lab. These values were used to calculate the sample sizes needed to achieve a significance level (α) of 0.05 and statistical power (1−β) of >0.8.

## Additional information

### Funding

| Funder | Grant reference number | Author |
|---|---|---|
| National Institutes of Health | K99HL155791 | Katelyn E Sadler |
| National Institutes of Health | R01NS070711 | Cheryl L Stucky |
| National Institutes of Health | R37NS108278 | Cheryl L Stucky |
| Advancing a Healthier Wisconsin | | Cheryl L Stucky |

The funders had no role in study design, data collection and interpretation, or the decision to submit the work for publication.

### Author contributions
Sydney Trask, Katelyn E Sadler, Conceptualization, Formal analysis, Investigation, Methodology, Visualization, Writing - original draft, Writing - review and editing; Jeffrey S Mogil, Conceptualization, Supervision, Writing - review and editing; Fred J Helmstetter, Cheryl L Stucky, Supervision, Writing - review and editing

### Author ORCIDs
Sydney Trask http://orcid.org/0000-0002-4396-5334
Cheryl L Stucky http://orcid.org/0000-0003-4966-6594
Katelyn E Sadler http://orcid.org/0000-0003-2078-3527

### Ethics
All protocols were in accordance with National Institute of Health guidelines and were approved by the Institutional Animal Care and Use Committee at the Medical College of Wisconsin (Milwaukee, WI; protocol #0383).

### Decision letter and Author response
Decision letter https://doi.org/10.7554/eLife.75283.sa1
Author response https://doi.org/10.7554/eLife.75283.sa2

## Additional files

### Supplementary files
• Transparent reporting form

### Data availability
All data generated or analyzed during this study are included in the manuscript.

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
