## [Editor Report]

This study covers a series of experiments designed to characterize conditioned pain processing using a novel animal model in which mechanical nociception (von Frey test) and writhing are assessed following exposure to contextual cues that have been paired with visceral pain (intraperitoneal acetic acid injection). The results reveal that such cues exert complex, dose- and sex-dependent effects on pain processing. These experiments address an important topic from a translational perspective, because learning is an important but understudied contributor to the human pain experience and because there is evidence for sex differences in human pain expression. These findings will be of broad interest to researchers across fields of associative learning, neuroscience, and pain research.

---

## [Decision Letter]

**Decision letter after peer review:**

Thank you for submitting your article "Contextual control of conditioned pain tolerance and endogenous analgesic systems: Evidence for sex-based differences in endogenous opioid engagement" for consideration by *eLife*. Your article has been reviewed by 3 peer reviewers, including Laura A Bradfield as Reviewing Editor and Reviewer #1, and the evaluation has been overseen by Michael Taffe as the Senior Editor. The following individuals involved in review of your submission have agreed to reveal their identity: Sean B Ostlund (Reviewer #2); Steve Negus (Reviewer #3).

Essential revisions:

Upon consultation with the reviewers, the essential revisions are seen as:

1) The inclusion of a control experiment in Figure 5, to investigate the effects of naloxone in vehicle-treated rats, as explained in more detail in the comments by Reviewer #1 and #3.

2) The evaluation of pain behaviours other than that of mechanical hypersensitivity, in line with the detailed comments provided by reviewers #2 and #3 is requested to determine the specificity (or indeed generality) of the behavioural responses to acetic acid injection.

3) Some rearrangement of the manuscript to improve logical flow is necessary. In particular, reviewers #1 and #2 note that they found Figure 1 confusing, along with other several detailed suggestions.

4) Some consideration of the qualitative vs. quantitative sex difference is necessary. Experimental exploration of this idea is encouraged but not essential, however, some thoughtful discussion of this issue is necessary.

*Reviewer #1 (Recommendations for the authors):*

Weakness 1:

– I suggest the authors make it clear and consistent between the text and figures as to which experiment is which (e.g. ascending dose experiment = Experiment 1, consistent dose experiment = Experiment 2, naloxone experiments = Experiments 3+ and so on).

– I also wasn't sure which animals received acetic acid injections prior to training, as shown in Figure 1. Was this all animals? Or is it just an illustration of what would happen if animals received acetic acid injections in the absence of context training? This needs to be made clearer.

– It could just be the way the figures converted to pdf or some other factor, but the figures I had were a bit blurry (except Figure 1) and the writing was a little on the small side.

– Where the naloxone experiments are reported and mice are no longer separated by sex, this needs to be signposted and a rationale given, particularly considering the large sex differences reported in equal sections. Moreover, it needs to be stated whether males and females are equally distributed among groups in Figures 4B, C, and E.

Weakness 2:

– I don't have a specific suggestion for this weakness, but I am curious as to whether the authors agree/disagree regarding whether the lack of difference from Day 1 vs. Day 4 in male mice in conditioned tolerance suggests that they did not develop this response? If not, why not?

Weakness 3:

– I suggest the authors temper their language here a little, e.g. "we did not find any evidence to suggest that the effects are mediated by corticosterone" rather than stating the effects were not mediated by corticosterone.

Weakness 4:

– The null effect in the vehicle group in Figure 4E relies on a very low sample size, n = 4. I always find it a little uncomfortable to conclude that a treatment did not have an effect based on such low numbers because the chances of a Type II error are so high. However, without any conditional pain tolerance in this group, I can see that it is very unlikely for naloxone to have any effect – effects are probably already at ceiling. I would suggest changing the wording in lines 314-315 to something like "we found no evidence for any effect of naloxone on referred mechanical…." Rather than saying naloxone had no effect.

Weakness 5:

– I suggest that the authors make their interpretation of the data reported in this section a bit clearer, particularly in light of their stated premise for these experiments (i.e. that it is unclear if the compensatory analgesic response was context-dependent or autonomic response to the acetic acid). To me it seems that the data in the males suggests the latter but I wasn't clear if that was the author's interpretation as well? And if so, what does this mean for the results in the rest of the manuscript?

– With regards to the females data shown in Figure 5E, either the authors would need to demonstrate that naloxone-mediated sensitivity is not present in the vehicle context in the absence of an acetic acid injection, or provide a good explanation as to why they are not including these data.

*Reviewer #2 (Recommendations for the authors):*

Below I outline a few specific issues and suggestions for strengthening the paper, though more general comments made in the public comments should also be considered.

1) Writhing and other responses to pain and pain-paired stimuli.

a. An important alternative explanation to consider is that behavioral effects of the pre-test treatment carried over into the test session in a way that interfered with von Frey testing. How would a general increase in writhing, recuperative (e.g., grooming), or locomotor behavior impact these measurements? Do the authors have any measurements of these behaviors during von Frey test sessions.

b. Did the authors measure writhing or other behaviors during the context exposure sessions? Understanding the time course of such outcomes would not only help address the alternative account just discussed, it could also provide great insight into the nature of the conditioned responses reported here. It is possible that the paired-context elicited a US-specific analgesic response which primarily countered the visceral pain related to the acid injection. This diminished visceral pain could itself explain the absence of mechanical allodynia, though it would not explain the conditioned analgesia observed in females in the final experiment. Alternatively, perhaps there was no attenuation of writhing or visceral pain in the paired context, which would strength the authors' conclusions and focus on the nociceptive touch. Did naloxone pretreatment diminish writhing and/or would such an effect be expected from the literature?

2) I believe the conceptual framework for the study and the model tested here could be presented more clearly. Figure 1 is particularly confusing until one has already read the methods, including the final experiment. Exactly what changes are needed depends on how the authors respond to my comment #1B, since I think there may be a need to emphasize the direct effects of the acid injection (visceral pain, writhing) and distinguish these from nociceptive touch assessment. I was also confused by the different "scenarios". Why is Scenario 1 associated with "examin(ing) pain sensitivity" when this is done in all experiments (if by pain sensitivity we mean von Frey testing)? Why is Scenario 2 association with "exam(ing) pain expectation" when this is done in all experiments (if by pain expectation we mean the conditioned effects of context cues)? If Scenario 2 is supposed to be linked to the last experiment (without pre-test acid injection), which it seems, why does it emphasize naloxone assessment, which was also done with the earlier design. I think the schematic tries to distinguish between some key differences in questions asked across studies but may cause more confusion that clarification. Here are a few suggestions. Consider using only the Scenario 1 example and filling in with more informative labels, like visceral pain and mechanical pain. The posture of the mouse alone does not make this clear. If the context only variation on the test is to be presented in "scenario 2", then I suggest leaving out the naloxone pill bottle (which again is not specific to this experiment and is represented in a confusing way). I would also use more objective terms for descriptors (e.g., Examine contextual modulation of visceral pain vs. mechanical pain). The term expectation here is somewhat loaded and would suggest a cognitive processes for some readers, which would be more compelling if there was evidence of pain type specificity or other indications of cognitive representation (e.g., inflation). And both the paired and unpaired contexts should be represented in any context-only Scenario 2 illustration.

3) Again, depending on how the authors respond to Comment #1B, I would be interested in a discussion of the somewhat paradoxical finding that pain-paired context cues support opposing conditioned pain responses in the final experiment. Presumably conditioned hypersensitivity and conditioned analgesia have different adaptive functions and are governed by different mechanisms. Elucidating the factors that differentially influence these CRs will likely be an important step in establishing this model and determining its impact on the field. It is also important to address why effects vary depending on whether or not visceral pain is administered prior to testing. Here are some items to consider:

a. Pain-specificity: I noted the possibility that this may relate to pain type/modality; i.e., visceral pain-paired cues may selectively block the 'expected' visceral pain, effectively canceling out the hypersensitivity to nociceptive touch that would have been induced by the pre-test injection. Perhaps such cues also act directly (in absence of visceral pain) on touch perception in order to increase vigilance, or sensitivity to outside threats.

b. Strength of learning or CR: Conditioned analgesia appears to depend on the strength of conditioning (or at least UCS intensity) in the first two experiments, so perhaps this also accounts for the sex difference in conditioned analgesia in the final experiment. But then where was the conditioned allodynia in the early experiments? Obscured by a floor effect?

c. Cue control: Is there reason to think that the visceral pain, itself, might be encoded as part of the training context, such that mice failed to fully generalize to the external context cues in the final experiment?

d. Timing: Is there any evidence that conditioned analgesia and allodynia are temporally and/or hierarchically organized, with the former preceding the latter. Perhaps this relates to strength of conditioning. If paired with a sufficiently strong pain UCS, context cues may elicit an immediate conditioned analgesic response with the persistence of this effect depending on associative strength. This might be followed by a conditioned state in which pain perception is heightened.

*Reviewer #3 (Recommendations for the authors):*

1. IP acid served as the US in this study, and the primary UR was mechanical hypersensitivity elicited by a second US (von Frey filaments) applied to a different anatomical target (hind paw). IP acid produces a number of other more direct URs (e.g. writhing, grimace, reduced locomotion, etc). l115 of Methods indicates that mice might have been video recorded during conditioning or test sessions. Were sessions recorded? Were these more direct behaviors scored either in real time or from recordings? Did US+CS pairing produce either "conditioned pain tolerance" to any of these more direct behaviors in the presence of US+CS (e.g. less writhing or grimace, more locomotion), or conditioned expression of these behaviors in the absence of the US (e.g. CS-induced expression of writhing, grimace, or reduced locomotion)? If these data are available, it would be helpful to include them. If they are not available, it would be helpful in Discussion to note that these other behaviors also function as IP acid-induced URs, they were not studied here, and the generality of US+CS pairing effects for these other behaviors remains to be examined.

2. An important finding in this study is the apparent higher potency of IP acid to elicit the mechanical hypersensitivity UR in females than males. Three comments on this finding. (a) First, how do sex differences in this UR relate to other published evidence for presence or absence of sex differences in other URs elicited by IP acid (e.g. Tansley 2019 reference, PMID: 20453868, PMID: 33622770)? Sex differences in expression of pain behaviors with other visceral pain models are mentioned (l468-470), but studies using IP acid are highly pertinent and should also be mentioned. My understanding is that these studies have generally observed a nonsignificant trend for higher expression by females of these other URs, including IP acid-induced thermal hypersensitivity in the Tansley 2019 paper. This suggests that the larger sex difference observed here for IP acid-induced mechanical hypersensitivity is unusual. (b) Second, how might this sex difference in potency of IP acid to induce acute hypersensitivity influence expression of the conditioned hypersensitivity expressed in males but not females in Figure 5? At present, this difference is described as a qualitative sex difference; however, an alternative explanation is that it is a quantitative potency difference in IP acid effects. Specifically, it is possible that a lower acid concentration or a smaller number of conditioning trials might produce conditioned hypersensitivity in females, or that a higher acid concentration or greater number of conditioning trials might engage an opioid-mediated resistance to conditioned hypersensitivity in males. This possibility could be explored experimentally as it was for conditioned pain tolerance, but it should at least be mentioned as a possibility in the Discussion. (c) Lastly, the first sentence of the Abstract suggests that associative learning mechanisms such as those studied here might contribute to the transition from acute to chronic pain, and the Discussion notes that chronic pain is more prevalent in women than men (l465). In this study, though, data in Figure 5 show conditioned expression of the pain behavior only in males. This greater expression of conditioned pain behavior in males appears inconsistent with the greater prevalence of chronic pain in women.

3. In Figure 5D and 5E, a control group should be included that receives vehicle on all three conditioning days followed by naloxone pretreatment to vehicle on the Day 4 test day. Current interpretation of data in Figure 5E assumes that naloxone would not produce mechanical hypersensitivity in vehicle trained mice, and inclusion of this control experiment would test this important prediction.

---

## [Author Response]

Essential revisions:Upon consultation with the reviewers, the essential revisions are seen as:1) The inclusion of a control experiment in Figure 5, to investigate the effects of naloxone in vehicle-treated rats, as explained in more detail in the comments by Reviewer #1 and #3.

This control experiment was added in as new Figure 4F. Over the course of 3 days, mice were repeatedly injected with PBS vehicle then exposed to a specific context for 60 minutes. On day 4, mice were pre-treated with naloxone or vehicle control prior to being placed into the training context. Hindpaw withdrawal thresholds were measured 45 min after placement into this context. Unlike the thresholds of animals trained with acid injections (see Figure 4E), hindpaw withdrawal thresholds of vehicle-trained mice were unchanged by naloxone administration, suggesting that endogenous opioid signaling does not influence pain sensitivity in environments that are not previously associated with pain.

2) The evaluation of pain behaviours other than that of mechanical hypersensitivity, in line with the detailed comments provided by reviewers #2 and #3 is requested to determine the specificity (or indeed generality) of the behavioural responses to acetic acid injection.

Writhing, a reflexive pain behavior initiated by acetic acid injection, was added into the following experiments: Figures 1E-G, 2E-G, and 3G-H. The inclusion of these data make it clear that the strength of the unconditioned stimulus affects the magnitude of conditioned analgesia. Although females exhibit acid-related hindpaw mechanical hypersensitivity (a phenomenon labeled ‘hyperalgesic descending control of nociception’ (DCN); see below) after training with low doses of acetic acid, they do not exhibit conditioned analgesia in the writhing test unless trained with high doses of acetic acid. Similarly, males, which do not exhibit hyperalgesic DCN unless trained with high doses of acetic acid, do not exhibit conditioned analgesia in the writhing component of any of the current experiments, suggesting that conditioning was not strong enough to evoke CS-induced analgesia. Additional discussion of stimulus generalization, a point raised by Reviewer 2, has been added to the discussion.

3) Some rearrangement of the manuscript to improve logical flow is necessary. In particular, reviewers #1 and #2 note that they found Figure 1 confusing, along with other several detailed suggestions.

Significant changes have been made to improve the flow of the manuscript. First, old Figure 1 was removed and replaced with a graphical summary (Figure 5). This new summary figure illustrates the experimental design and behavioral results obtained in each sex for all conditioning paradigms used in the manuscript. Second, and at the request of public comments on BioRxiv, we have updated terminology used throughout the manuscript as follows: while the appropriate translation of Pavlov’s original Russian text, “conditional” and “unconditional” have been replaced with “conditioned” and “unconditioned”; “pain tolerance” has been replaced with “analgesia”; the term “anti-DNIC” has been replaced with “hyperalgesic descending control of nociception (DCN)”. This last change adheres to the terminology suggestions put forth by Bannister et al., (2021) Pain, which clearly distinguishes between the phenomena known as “conditioned pain modulation”, “diffuse noxious inhibitory controls”, and “descending control of nociception”. Third, additional details were added to the methods section to make our experimental timeline explicitly clear.

4) Some consideration of the qualitative vs. quantitative sex difference is necessary. Experimental exploration of this idea is encouraged but not essential, however, some thoughtful discussion of this issue is necessary.

This topic is now specifically addressed in the discussion. References for manuscripts that have used acetic acid as a conditioning stimulus are also included throughout the text, as are day-by-day hindpaw withdrawal thresholds (Supp. Figure 2-1) for all animals included in Figure 2 experiments. The addition of these data and the writhing behaviors described above suggest that there are quantitative sex differences in the current datasets, and the possibility of qualitative differences when UCS are not present on test day (i.e., results in Figure 4).